# Automated Barometric Chamber for Entomology Experiments: Arthropods' Behavior and Insect-Plant Interactions

**Camila M. Costa** [1], **Antonio P. Camargo** [2],*[ID], **Eric Alberto da Silva** [1] **and José Maurício S. Bento** [1][ID]

1    Escola Superior de Agricultura 'Luiz de Queiroz' (ESALQ), Universidade de São Paulo (USP),
     Piracicaba 13418-900, SP, Brazil; camilamcostaa@gmail.com (C.M.C.); ericsilva@usp.br (E.A.d.S.);
     jmsbento@usp.br (J.M.S.B.)
2    Faculdade de Engenharia Agrícola (FEAGRI), Universidade Estadual de Campinas (UNICAMP),
     Campinas 13083-875, SP, Brazil
*    Correspondence: apcpires@unicamp.br

**Featured Application: An automated barometric pressure chamber for entomology research was designed; the system can change the barometric pressure by ±15 hPa from the local value; the barometric pressure is maintained with a stability of ±0.1 hPa; the system allows the pressure to be changed slowly and according to linear ramps; applications are related to arthropods' behavior and arthropod–plant interactions.**

**Abstract:** Insect behaviors, such as flying, oviposition, parasitism, mating/calling, response to semiochemicals, and others, might be influenced by barometric pressure fluctuations. Abiotic factors controlled in the laboratory facilitate the observation of particularities related to development, behavior, and/or habits of arthropods and plants and their interactions. This study aimed to design an automated barometric chamber for research on arthropod behaviors and insect–plant interactions in the laboratory. The barometric chamber is a transparent box equipped with a single-board computer. An air pump and two proportional solenoid valves were used as actuators to control the air flow, while barometric pressure, air humidity, and temperature sensors were used to monitor the conditions within the chamber. A graphical user interface to operate the barometric chamber was developed to run in a web browser. The barometric chamber was designed to allow the barometric pressure to be changed by up to 15 hPa with respect to the local barometric pressure. In addition, the control system makes it possible to set the rise/fall time (ramp) corresponding to the duration in which a change of pressure will be conditioned. Short- and long-term evaluations demonstrated that the control system can assure pressure stability of ±0.1 hPa with respect to the setpoint value. For demonstration purposes, two experiments were carried out to evaluate the influence of barometric pressure on the feeding activity of *Euschistus heros* and *Diabrotica speciosa*. For *E. heros*, the number of stylet sheath was significantly increased under high pressure conditions compared to the low pressure. However, for *D. speciosa*, there was no statistical difference in leaf consumption at the evaluated testing conditions.

**Keywords:** barometric pressure; instrumentation; insects' behavior; abiotic factors

## 1. Introduction

The environment inhabited by an organism may influence and modify its way of living [1]. Environmental conditions are composed of abiotic factors such as wind, rain, soil components, radiation, temperature, and atmospheric pressure [2]. Abiotic factors can be controlled in the laboratory to facilitate the observation of particularities related to the development, behavior, and/or habits of insects and plants [3,4].

Atmospheric pressure is important since large-scale weather changes are related to changes in pressure systems. Atmospheric or barometric pressure is the force exerted by atmospheric air on any surface [5]. Average atmospheric pressure at sea level is 1013 hPa [6];

however, there are variations which can be divided into regular and irregular. Regular variations are related to changes in altitude—the pressure decreases at higher elevations and vice versa—and to semi-daily variations due to heating and cooling of air [5]. Above sea level, atmospheric pressure drops about 10 hPa per 100 m of increased altitude [7,8]. Irregular variations are related to weather changes due to the passage of pressure systems [5]. Low-pressure systems are called cyclones and tend to produce storms. In these systems, the hot air near the surface moves inward, resulting in a convergent flow. Since rising air results in the formation of clouds and precipitation, a low-pressure system is usually associated with adverse weather conditions involving strong winds and storms [5]. In a high-pressure system, called an anticyclone, there is divergent flow near the earth's surface. As the descending air is compressed and heated, the formation of clouds and precipitation is less likely to occur [7].

Recent studies have shown that adverse weather conditions, often associated with barometric pressure changes, can affect reproductive success, food availability, and/or habitat adequacy [9–22]. Insect behaviors, such as flying, oviposition, parasitism, mating/calling, response to semiochemicals, and others, might be influenced by barometric pressure. Parasitoid females of *Trichogramma pretiosum* (Riley) and *T. evanescens* (Westwood) (Hymenoptera: Trichogrammatidae) immediately diminished flight initiation at the start of rapid pressure changes, probably as an attempt to avoid risk of death or uncontrolled flight during upcoming adverse weather conditions [9]. Flight initiation of the psyllid *Diaphorina citri* changed in response to variations in barometric pressure. *D. citri* dispersed more as barometric pressure increased and less as it decreased [10]. The parasitoid *Aphidius nigripes* (Ashmead) (Hymenoptera: Aphidiinae) also showed a decrease in flight initiation when the barometric pressure varied by 5 hPa, and the chemical communication of this parasitoid was also affected by variations in barometric pressure [11]. A similar pattern was observed in the curculionid *Conotrachelus nenuphar* (Herbst) (Coleoptera: Curculionidae), whose discrimination of host plant volatiles was affected [12]. An increase in daily response to infochemicals by *Cotesia glomerata* (L.) (Hymenoptera: Braconidae) was associated with a daily increase in barometric pressure [13]. Regarding the oviposition behavior of *Pieris rapae* (L.) (Lepidoptera: Pieridae), *Conotrachelus nenuphar* (Herbst) (Coleoptera: Curculionidae), and *Leptopilina heterotoma* (Thomson) (Hymenoptera: Figitidae), a higher number of eggs were laid under low barometric pressure [12,14,15]. Mating and calling behaviors were also affected by barometric pressure in *Pseudaletia unipuncta* (Lepidoptera: Noctuidae) and *Macrosiphum euphorbiae* (Hemiptera: Aphididae) [16]. In *Frankliniella schultzei* (Trybom) (Thysanoptera: Thripidae), only rapid pressure drops, like cyclones, resulted in a lower number of matings [17]. In *Diaphorina citri* (Kuwayama) (Hemiptera: Liviidae), mating was reduced due to variations in barometric pressure [18]. Several other studies show the effect of barometric pressure on shelter-seeking behavior, adult emergence, foraging and defense by social insects, behavioral learning, and others [17,19–21]. Plants also seems to present mechanisms of perceiving barometric pressure changes [22] but studies regarding this are scarce.

Laboratory research usually seeks correlations between behavioral responses and variations in barometric pressure under natural conditions. Only a few studies have manipulated and controlled barometric pressure using a barometric chamber [9,10,16,17,23,24] and none of them presented a barometric chamber equipped with a control system as accurate as the one proposed in this research.

This study aimed to design an automated barometric chamber to carry out research on arthropod behaviors and insect–plant interactions in the laboratory. For demonstration purposes, two experiments were carried out to evaluate the influence of barometric pressure on the feeding activity of *Euschistus heros* (Fabr.) (Hemiptera: Pentatomidae) and *Diabrotica speciosa* (Germ.) (Coleoptera: Chrysomelidae).

## 2. Materials and Methods

The automated pressure chamber was designed for research purposes and installed at the National Institute of Science and Technology (INCT)—Semiochemicals in Agriculture (Piracicaba, SP, Brazil). The barometric chamber described hereafter was developed for entomology experiments in which the barometric pressure within the chamber might reproduce real atmospheric pressure fluctuations.

### 2.1. Barometric Pressure Variations in São Paulo State, Brazil

The barometric chamber was designed to reproduce small barometric pressure variations associated with weather changes in daytime periods. Therefore, before designing the barometric chamber and its control system, weather data from Sao Paulo state were analyzed to define the operation limits of the chamber.

Figure 1 summarizes five years of barometric pressure data for some cities in São Paulo state, Brazil, where hourly barometric pressure data were available. Data were obtained from the Brazilian National Institute of Meteorology (INMET) database and from the weather station available at ESALQ/USP.

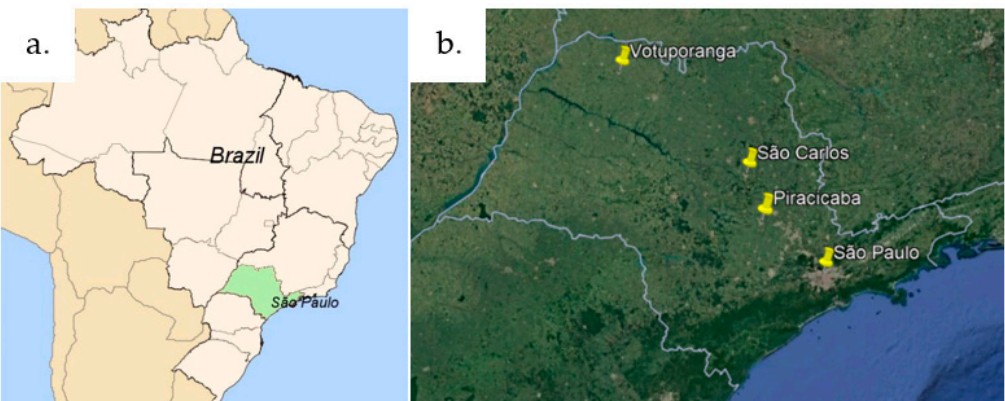

| Cities | Altitude (m) | Barometric pressure (hPa) | | | | |
|---|---|---|---|---|---|---|
| | | Min | Avg | Max | Max – Avg | Avg – Min |
| Piracicaba | 546.0 | 938.9 | 951.1 | 964.0 | 12.9 | 12.2 |
| São Carlos | 859.3 | 908.6 | 920.1 | 932.2 | 12.1 | 11.5 |
| São Paulo | 785.6 | 914.7 | 927.3 | 940.3 | 13.0 | 12.6 |
| Votuporanga | 510.4 | 949.0 | 960.4 | 973.4 | 13.0 | 11.4 |

**Figure 1.** Minimum, average, maximum, and deviations from the average values of barometric pressure (hPa) in cities of São Paulo state, Brazil, from January 2017 to December 2021 ((**a**): location of São Paulo state in Brazil; (**b**): location of the cities where barometric pressure data were analyzed; (**c**): data of barometric pressure).

Figure 2 shows barometric pressure data gathered every hour for Piracicaba (São Paulo state, Brazil) for five years (weather station at ESALQ/USP).

Based on the deviations from the average (Figures 1 and 2), barometric pressure variations in São Paulo state did not exceed 13.0 hPa and, consequently, variations higher than that are not expected to occur in São Paulo state unless extreme weather events, such as hurricanes, occur. As a matter of curiosity concerning extreme weather events, Ref. [7] reports that barometric pressure may drop by 60 hPa from the outer edge of a hurricane to its center.

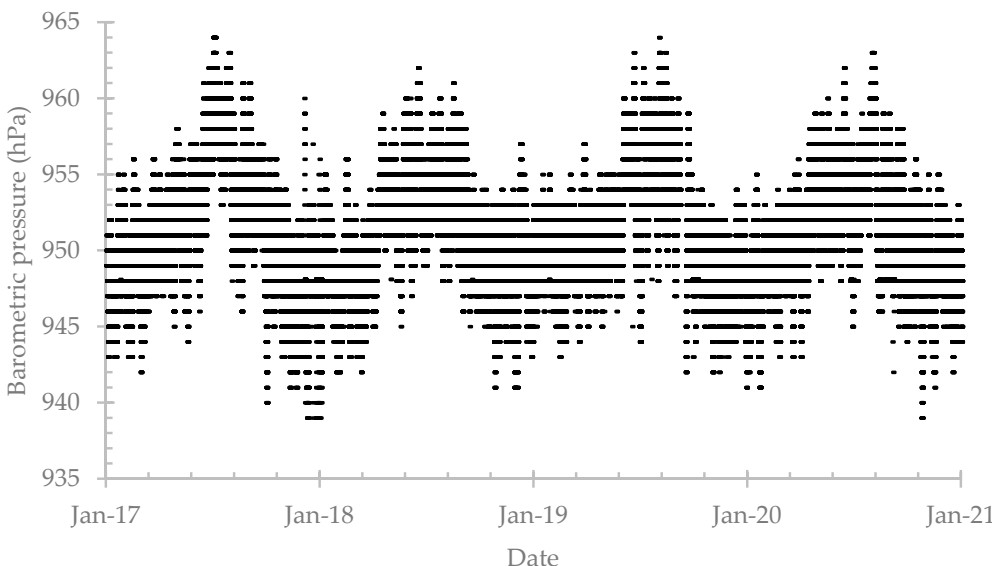

**Figure 2.** Range of values and natural fluctuations of barometric pressure (hPa) at Piracicaba (São Paulo state, Brazil), from January 2017 to December 2021, used as a historical dataset for defining the operational requirements of the barometric chamber.

Variations in barometric pressure are expected to occur very slowly in nature except in the presence of extreme weather events [5,7]. Based on the historical dataset analyzed, we determined that the barometric chamber must allow the barometric pressure to be changed by up to ±15 hPa with respect to the current barometric pressure. In addition, since variations in barometric pressure occur very slowly in nature, the control system must make it possible to set the rise/fall time corresponding to the duration in which a change of pressure will be conditioned.

### 2.2. The Barometric Chamber and Its Components

The barometric chamber is an acrylic box 900 mm long, 700 mm high, and 400 mm deep made of acrylic sheets with a thickness of 8 mm (Figure 3). All surfaces of the box are transparent to allow the filming of arthropod behavior or interactions between arthropods and plants. The front wall of the chamber works as a lid that is attached to the chamber with bolts. Between the lid and the chamber, 6-mm sealing tape was installed to minimize air leakage. In one of the lateral walls, there are entry points for the inlet/outlet of air and wire connections for sensors installed inside the chamber.

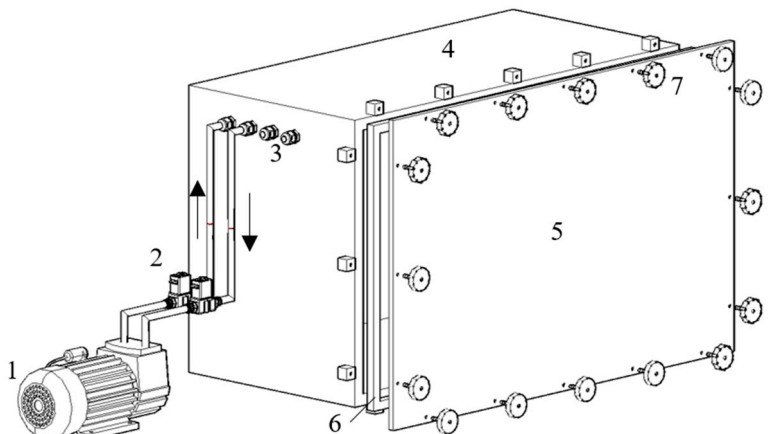

**Figure 3.** Schematic of the barometric chamber components: (1) pressure/vacuum air pump; (2) pressure and vacuum proportional valves; (3) wire connections; (4) barometric chamber box; (5) barometric chamber lid; (6) sealing tape; (7) bolts.

A Millipore model WP61 vacuum/pressure pump is used to increase or decrease the pressure in the barometric chamber. The air pump is a continuously running constant air flow type unit for use with laboratory equipment [25]. The air pump has a positive pressure port and a negative one. According to the manufacturer, the pump has the following specifications: a maximum vacuum of 67.7 kPa at a flow rate of 4 L min$^{-1}$ and a maximum pressure of 140 kPa at a flow rate of 16 L min$^{-1}$. Values of both pressure and vacuum decrease as the flow rate increases, as shown in the curve of pressure versus flow rate that characterizes the pump [25]. The air pump power supply is 127 V/60 Hz.

Air flow control within the chamber is achieved using two proportional solenoid valves, one operating as the pressure control valve and the other as the vacuum control valve. The valves were manufactured by ASCO and have the following specifications: Series 202, model Posiflow, directly operated, $\frac{1}{4}''$ pipe size, 4 mm orifice size, two-way, and normally closed (catalog number SCG202A004V [26]). The flow across these valves is proportional to an input control signal that ranges from 0 to 24 Vdc and may be obtained by pulse width modulation (PWM) at 300 Hz frequency.

The barometric pressure inside the chamber was monitored by a Young model 61302V (Young, Traverse City, MI, USA) barometric pressure sensor with a resolution of 0.01 hPa [27]. The sensor was set to operate in the range from 850 to 1050 hPa in which voltage output varies from 0 to 5000 mV. Equation (1) is given by the manufacturer to convert the analogue signal into barometric pressure.

$$hPa = 0.04 \, mV + 850 \tag{1}$$

Temperature and air humidity within the chamber were monitored by a DHT22 sensor (accuracy: temperature = 0.5 °C, air humidity = 2%). The chamber itself does not have temperature control but is installed inside a room where an air conditioning system keeps the air temperature steady. During all experiments reported, the room temperature was maintained at 25 ± 1 °C.

## 2.3. Control System

The chamber must allow the barometric pressure to be changed by up to ±15 hPa with respect to the current ambient barometric pressure. The control system must allow a given setpoint of barometric pressure to be reached and maintained but must also comply with a specific rise/fall time (i.e., a ramp) defined by the user. According to the requirements defined by the entomology team, the pressure control system was designed for experiments consisting of four stages. In stage 1, the barometric pressure is slowly adjusted to the 'acclimation pressure' complying with a rise or fall time defined by the user; in stage 2, the acclimation pressure is held steady for the duration defined by the user in order to enable acclimation of specimens during the experiment; in stage 3, the barometric pressure is increased or decreased to the 'test pressure' complying with a rise or fall time defined by the user; lastly, in stage 4, the test pressure is held steady until the experiment is finished.

For pressures higher than the local barometric pressure, the pressure control valve allows the control of air flow into the chamber. On the other hand, the vacuum control valve is used to obtain barometric pressures lower than the local values.

A Proportional-Integral (PI) controller is employed in all stages to control the barometric pressure. During stages 2 and 4, the control system maintains a fixed setpoint of barometric pressure. During stages 1 and 3, when control must comply with the rise or fall time, the setpoint is set in steps of 0.2 hPa following a linear function. Based on the input values defined by the user, the control system defines a linear equation for the setpoint as a function of time that allows calculation of the values of the setpoint every time the controller output is computed.

The PI controller was tuned empirically through several attempts in which the controller parameters were tested until a satisfactory performance was achieved for the entomology application. When the vacuum control valve is in use, the barometric pressure becomes inversely proportional to the aperture of the vacuum valve and consequently the

controller action is reversed; hence its parameters become negative. The PI controller reads the barometric pressure transmitter and computes new output values every 0.2 s when the control routine is enabled.

### 2.4. Hardware

A Raspberry Pi 3 B+ single-board computer was employed as the main component of the barometric chamber circuitry (Figure 4). Analog signals from the barometric pressure sensor were acquired using an ADS1115, which is an analog-to-digital converter with 16-bit resolution that incorporates a programmable gain amplifier (PGA). The I2C interface of the Raspberry Pi was enabled to interface with the ADS1115 and the gain of the PGA was set to 1. Air humidity and temperature inside the chamber were monitored using a DHT22, which is a low-cost module connected to the Raspberry Pi over the 1-Wire protocol interface. A relay circuit was used to switch the air pump. A Raspberry GPIO (i.e., general purpose input/output) pin was connected to the base of an NPN transistor (model BC548); this pin switches an electromechanical 12-V relay (127 V/10 A) that enables the pump. In addition, a resistor-capacitor snubber circuit was installed in parallel with the load (i.e., the pump) to suppress voltage surges while switching the air pump on/off. The two proportional solenoid valves were controlled by PWM at 300 Hz, as recommended by the valve manufacturer. PWM signals were generated by software (10-bit resolution) using two GPIOs of the Raspberry. NPN transistors (model TIP122) were used to power the valves. 1N4007 diodes were used as flyback diodes connected in parallel with the valves and with the relay inductor to suppress the sudden voltage spike that might occur while switching the current across these components. Finally, a 10.1" LCD touch screen was connected to the HDMI port of the Raspberry Pi to enable the user to operate the pressure chamber over a graphical user interface (GUI).

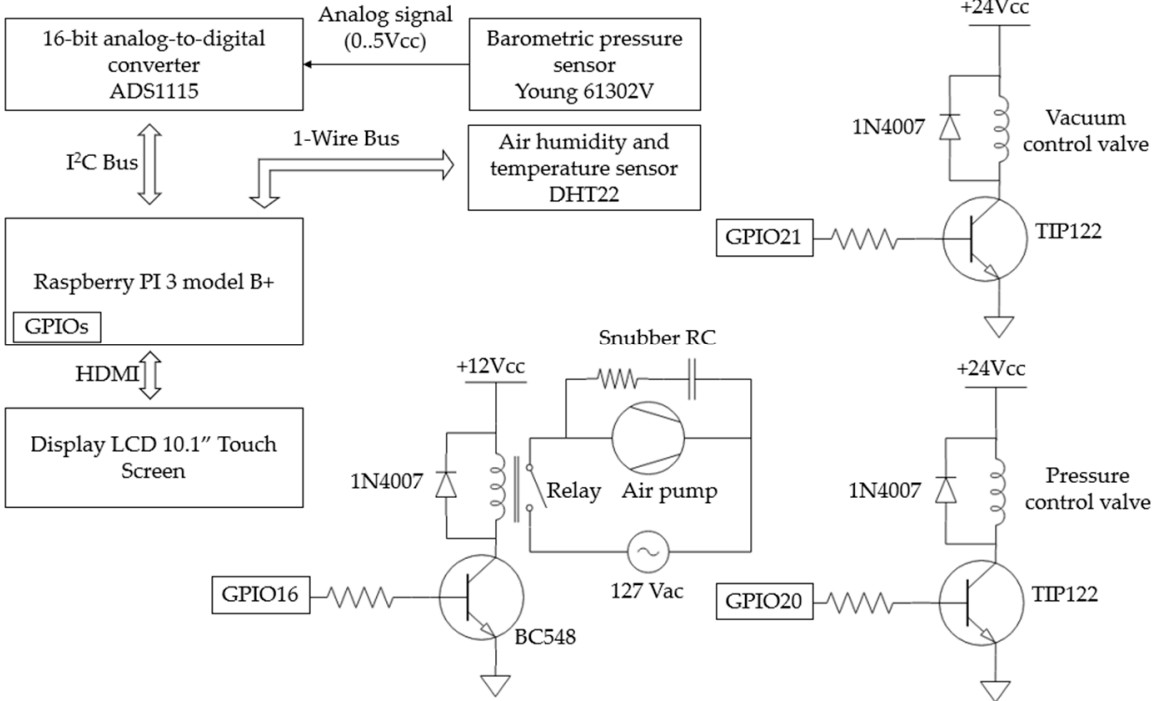

**Figure 4.** Diagram that represents the components of the electrical circuit responsible for the barometric chamber pressure control.

### 2.5. Application Architecture

A GUI to operate the barometric chamber was developed to run in a web browser in order to enable the user to control the chamber via the touch screen display or remotely (Figure 5). The application developed to automate the control and monitoring routines of

the pressure chamber consist of two programs coded in Python, running simultaneously: (1) webApp.py and (2) controller.py. The program webApp.py launches Flask, which is a Python framework used for developing web applications. Basically, when webApp.py is launched, Flask activates a web server in the Raspberry PI, loads all static files and templates (HTML, CSS, JavaScript, and images), and handles HTML requests by routing mechanisms processed by Python methods. The template and static files necessary to render the web pages were developed using HTML, CSS, and JavaScript. The web pages allow the user to set parameters required to operate the chamber; configure, start, and stop experiments; monitor variables inside the chamber; and see charts and generate reports. Dynamic data shown on the web pages are retrieved from a MySQL database. JavaScript/AJAX methods generate HTML requests mapped by Python. Python methods execute the requested tasks in the database and send HTML responses back to the client. When an experiment is running, charts and fields in the GUI are updated every 5 s by sending requests to the database. When the user sets a variable in a web page, a field in the database is updated. Changes in the GUI cause updates in the database 'control' table, but there is no direct change in GPIOs due to requests made by the user over the web pages.

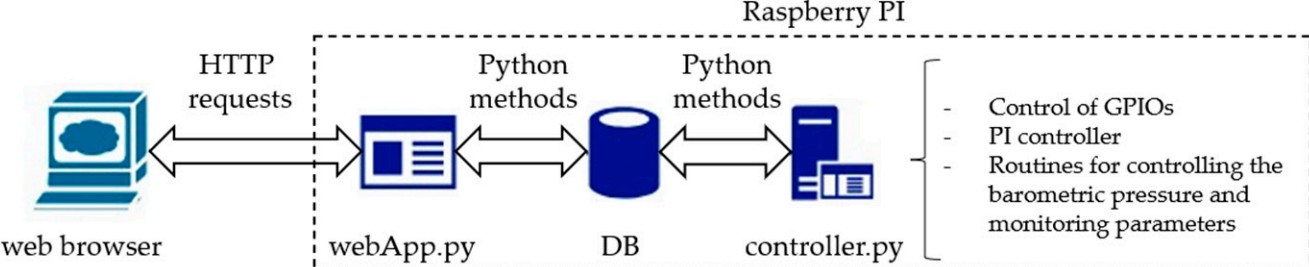

**Figure 5.** Application architecture developed to automate the control and monitoring routines of the pressure chamber.

The controller.py is the application in charge of synchronizing the database 'control' table and the GPIOs, as well as launching routines for controlling the barometric pressure and for monitoring variables inside the chamber. The database (DB) has only two tables: (1) 'control' and (2) 'records'. The 'control' table has several fields but a single record. Only update operations are allowed in the 'control' table and such operations may occur due to requests from controller.py or webApp.py. The 'records' table also has several fields and is used to save data gathered during experiments carried out in the barometric chamber. Data in this table are used to generate charts and reports of experiments. The controller.py also has the algorithm in charge of controlling the barometric pressure within the chamber. There is a routine used for ramp stages (i.e., stages 1 and 3) and another for steady stages (i.e., stages 2 and 4). Both routines implement a PI controller that reads the barometer and computes new output values every 0.2 s, as already mentioned in the control system.

The application architecture was designed as described to allow the browser to be closed while running an experiment in the barometric chamber and to allow monitoring of the experiments remotely over the internet. Once an experiment has been launched by the user over the browser, all control tasks are handled on the server side by the application controller.py. When an experiment is running, the database is continuously updated, whether or not the browser is opened. In this way, when a user accesses the web pages, information about the ongoing or previous experiments is retrieved from the database.

*2.6. Entomology Experiments: Feeding Activity of Insects*

For demonstration purposes, two experiments were carried out to evaluate the influence of barometric pressure on feeding activity of *Euschistus heros* (Fabr.) (Hemiptera: Pentatomidae) and *Diabrotica speciosa* (Germ.) (Coleoptera: Chrysomelidae).

The barometric chamber was in a temperature-controlled room kept at 24 ± 2 °C and 60 ± 10% RH, 12L:12D, and under 2500 lux of artificial light. At the beginning of each experiment, the insects were individualized in four release devices installed inside the chamber (Figure 6), the pressure chamber was sealed, and the target acclimation pressure (950 hPa) was programmed to be reached in 1 h. All abiotic conditions were kept constant, except pressure, which was controlled according to the following treatments: (1) stable (S)—insects were kept at 950 hPa during the whole experiment; (2) high (H)—insects were kept at 950 hPa for acclimation during 3 h and then pressure was gradually increased to 958 hPa over the next 6 h; (3) low (L)—insects were held at 950 hPa for acclimation during 3 h and then the pressure was decreased to 942 hPa over the next 6 h. After the 10 h period, the barometric pressure was kept steady and insects were released to feed during the next 12 h. As the feeding of *D. speciosa* and *E. heros* is a diurnal activity, the 10 h period of pressure manipulation occurred during the scotophase.

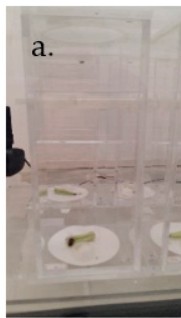 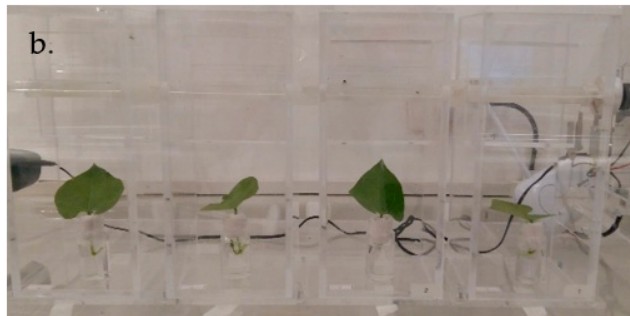

**Figure 6.** Release devices installed inside the chamber: (**a**) segments of bean pods offered to *E. heros* adults; (**b**) bean leaves offered to *D. speciosa* adults.

*E. heros* assays were carried out using 8–12 day old adults. To evaluate feeding activity, a bean pod section, approximately 3.5 cm long, was offered with distilled water in saturated cotton on a circular plastic container. After the 12 h feeding period, each pod was removed and subjected to the 1% acid fuchsin staining test for 15 min [28]. The pods were then washed in water and the number of stained stylet sheaths was counted using a stereoscopic microscope.

*D. speciosa* assays were carried out using 7–10 day old adults. To evaluate feeding activity, a leaf of bean *Phaseolus vulgaris* (cv. Carioca) was provided in a container with water. The initial leaf area was measured prior to the experiment. After the 12 h feeding period, the leaves were scanned, and the images were imported for Image J 1.44P software to calculate the consumed leaf area.

Eight replications were performed with four insects (subsamples) (N = 32) to evaluate *D. speciosa* and *E. heros* feeding activity. A linear mixed-effects model was fitted to the 'leaf area consumed' and a Poisson generalized linear mixed-effects model was fitted to 'number of stained stylet sheath', including the fixed effects of pressure and a random effect per each sample, given observations within the same sample are correlated. Goodness-of-fit was assessed using half-normal plots with simulation envelopes. The significance of the pressure effect was assessed through a likelihood-ratio (LR) test and multiple comparisons were performed by obtaining the 95% confidence intervals for the linear predictors.

## 3. Results and Discussion

### 3.1. The Barometric Chamber

Figure 7 shows the barometric chamber installed in the laboratory. The air pump and the control valves were installed inside a cabinet to minimize noise and vibration in the room. From the initial web page, the user can access functionalities to set up experiments, monitor sensor readings, manually control the actuators of the system, start and monitor experiments, and generate reports.

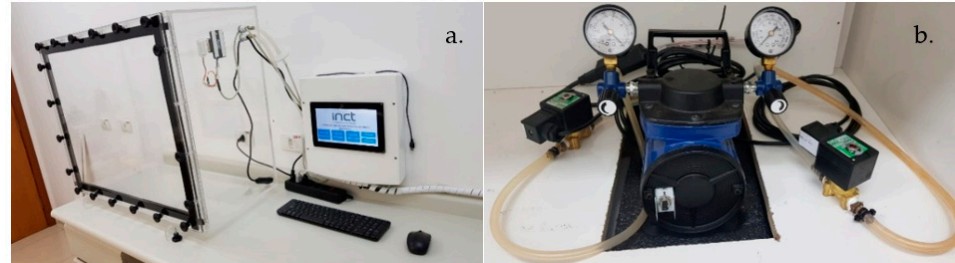

**Figure 7.** Barometric chamber built for entomology experiments (INCT Semiochemicals in Agriculture): (**a**) chamber and electronic control system; (**b**) air pump and proportional solenoid valves.

*3.2. Pressure and Vacuum Limits*

Figure 8 presents the maximum barometric pressure variation while pressuring and depressurizing the chamber. During 20 min of testing for the positive pressure, the barometric pressure increased by 19.63 hPa, starting from an initial value of 940.41 hPa. For the initial pressure of 940.36 hPa, barometric pressure decreased by 47.15 hPa while evaluating the negative pressure. Both positive and negative pressures are sufficient for the proposed entomology experiments, which require pressure variations of ±15 hPa. The observed differences in pressure capacity while pressurizing and depressurizing the chamber are related to the operational characteristics of the air pump, as shown on the datasheet [25]. Higher values of pressure variation could be obtained by selecting an air pump of higher flow capacity. The control system (controller.py) has a safety routine, in which it continuously monitors the barometric pressure and turns off the air pump if the barometric pressure is beyond the stated safety limits, ranging from 850 to 1050 hPa in this application. In addition, if higher values of positive/negative pressure are required by the application, the air pump can be replaced by separate devices dedicated to pressurizing (e.g., air compressor) and depressurizing (e.g., vacuum pump) the chamber. Values beyond the range of 850 to 1050 hPa might be achieved by the apparatus but, as far as we understand, there are no practical reasons to run experiments beyond the mentioned range. In addition, beyond the stated limits the chamber should be redesigned and built to support the operating pressures.

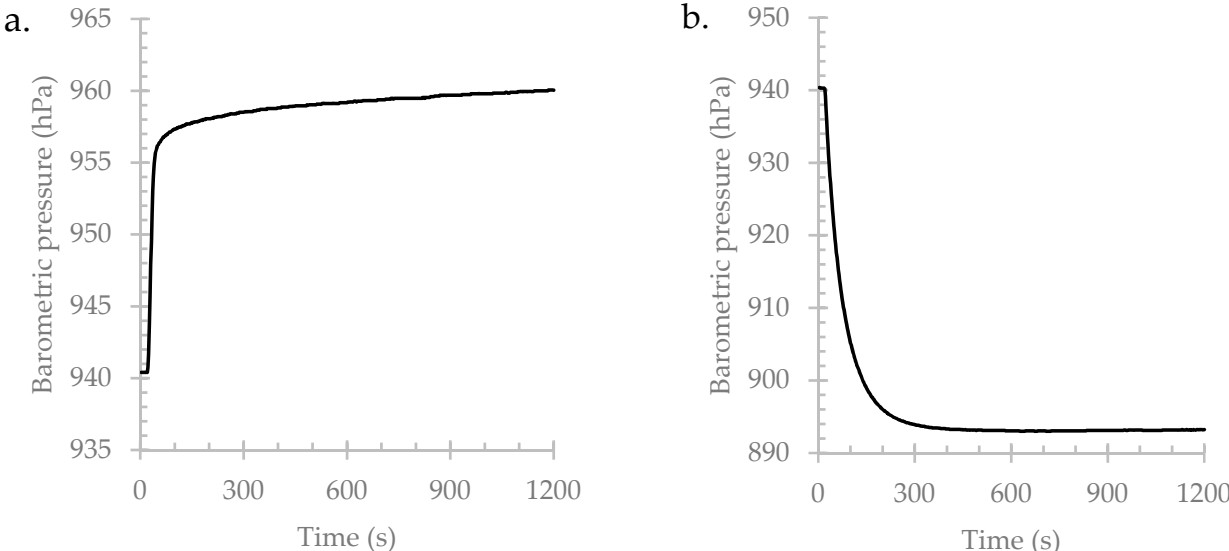

**Figure 8.** Limits of pressure (hPa) inside the chamber: (**a**) maximum positive pressure; (**b**) maximum vacuum. Lines indicate the maximum variation in barometric pressure that can be achieve while pressurizing and depressurizing the chamber in its current setup.

For comparison, the computer-controlled barometric chamber designed by [24] employed compressed air and a vacuum pump as the air source to change the barometric pressure and their system was designed to operate in the range from 950 to 1050 hPa.

### 3.3. Short- and Long-Term Tests

Figure 9 shows the results of short-term tests run to demonstrate that the chamber works as expected. Test 1 (Figure 9a,b) consisted of the following stages: (1) stage 1—the barometric pressure was gradually increased to 945 hPa in 10 min; (2) stage 2—a pressure of 945 hPa was maintained for 10 min; (3) stage 3—the barometric pressure was gradually increased to 955 hPa in 60 min; (4) stage 4—a pressure of 955 hPa was maintained for 10 min. Test 2 (Figure 9c,d) consisted of the following stages: (1) stage 1—the barometric pressure was gradually decreased to 935 hPa in 10 min; (2) stage 2—a pressure of 935 hPa was maintained for 10 min; (3) stage 3—the barometric pressure was gradually decreased to 925 hPa in 60 min; (4) stage 4—a pressure of 925 hPa was maintained for 10 min.

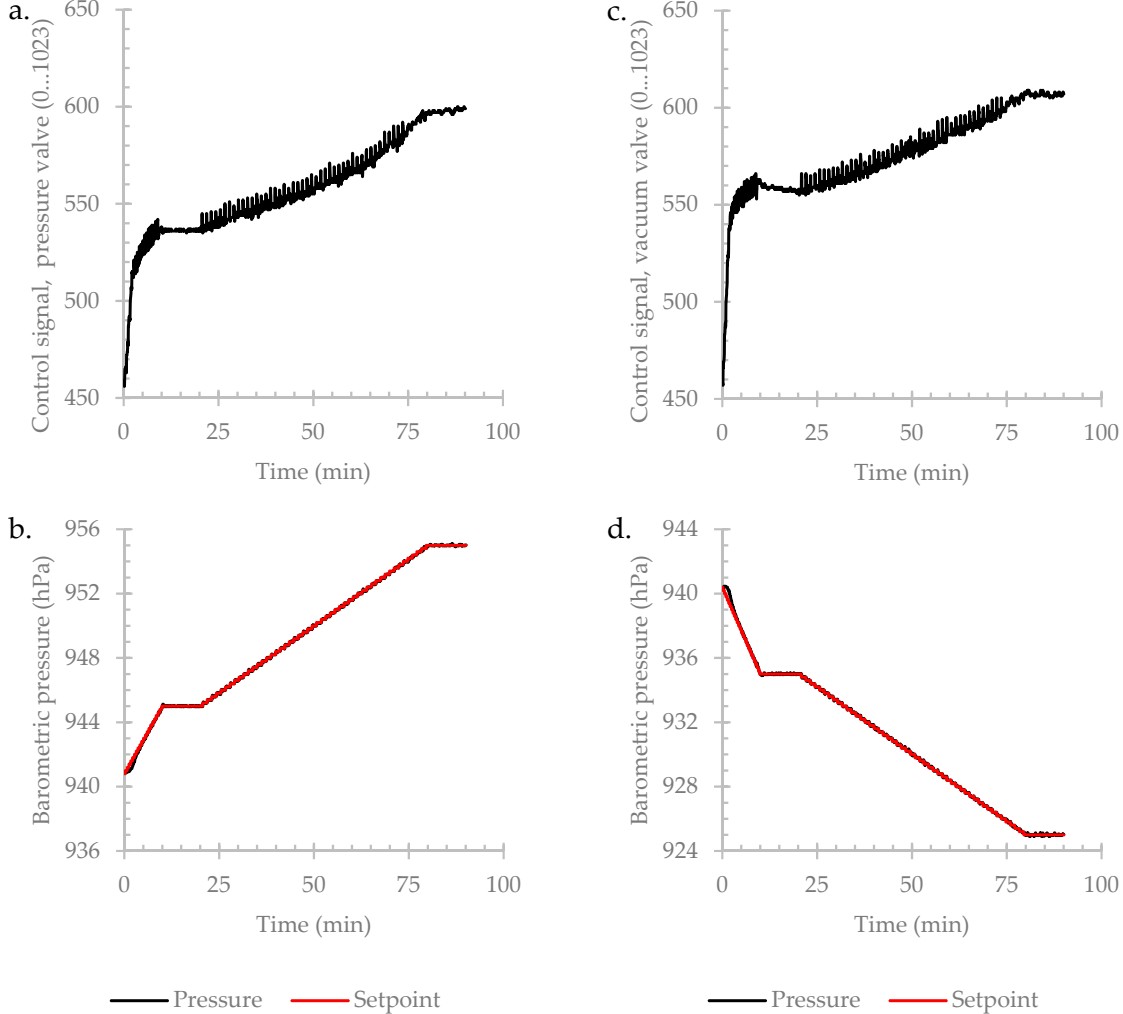

**Figure 9.** Results of short-term tests run to demonstrate that the chamber works as expected: (**a**) test 1—control signal fluctuations while controlling the pressure valve; (**b**) test 1—rising pressure within the barometric chamber; (**c**) test 2—control signal fluctuations while controlling the vacuum valve; (**d**) test 2—decreasing pressure within the barometric chamber.

Figure 10 shows the results of long-term tests run to demonstrate that the chamber works as expected. Test 3 (Figure 10a,b) consisted of the following stages: (1) stage 1—the barometric pressure was gradually increased to 945 hPa in 60 min; (2) stage 2—a pressure

of 945 hPa was maintained for 20 min; (3) stage 3—the barometric pressure was gradually increased to 955 hPa in 180 min; (4) stage 4—a pressure of 955 hPa was maintained for 20 min. Test 4 (Figure 10c,d) consisted of the following stages: (1) stage 1—the barometric pressure was gradually decreased to 935 hPa in 60 min; (2) stage 2—a pressure of 935 hPa was maintained for 20 min; (3) stage 3—the barometric pressure was gradually decreased to 925 hPa in 180 min; (4) stage 4—a pressure of 925 hPa was maintained for 20 min.

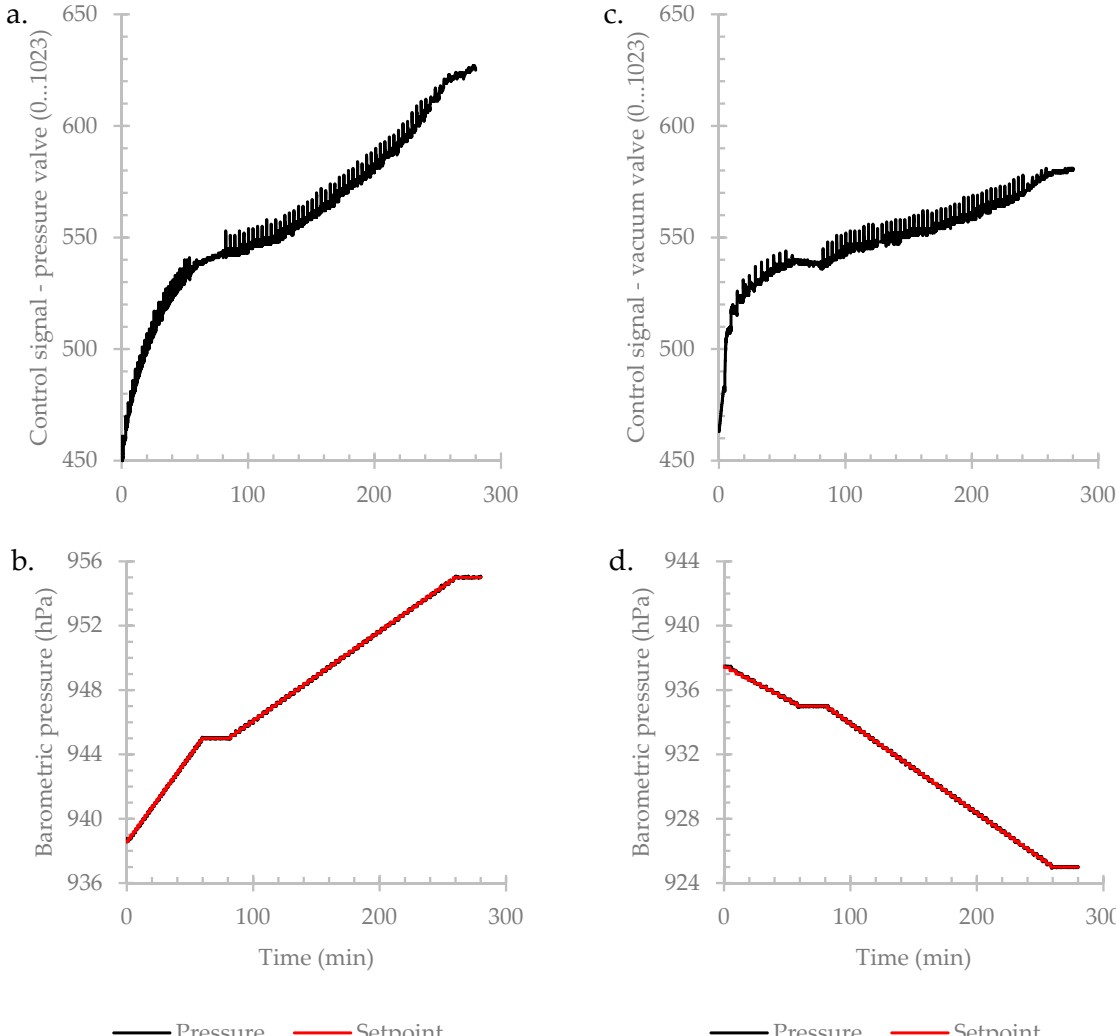

**Figure 10.** Results of long-term tests run to demonstrate that the chamber works as expected: (**a**) test 3—control signal fluctuations while controlling the pressure valve; (**b**) test 3—rising pressure within the barometric chamber; (**c**) test 4—control signal fluctuations while controlling the vacuum valve; (**d**) test 4—decreasing pressure within the barometric chamber.

The control signal applied to the valves ranges from 0 to 1023 (10-bit value). Experimentally, we observed that values from 0 to 450 correspond to a dead band in which there is no pressure response inside the chamber. Hence, minimum values of the control signal were set in 450 to achieve a faster response while controlling pressure. The barometric pressure results obtained in the short-term and long-term experiments (Figures 9 and 10) demonstrate that the control system operated properly, complying with the setpoint values and with the specifications for rise/fall time. The results from test stages 1 and 3 (i.e., ramps) indicated linearity while varying the barometric pressure. The linear function in the ramp stages was chosen by the entomology team as a good way to simulate real fluctuations, but other mathematical models could be implemented to change pressure in these stages. Likewise, the experiments comprise four stages due to the requirements specified

by the entomology team, but other strategies/methodologies could be implemented in the algorithm.

Table 1 shows descriptive statistics of the stages of the steady setpoint (i.e., stages 2 and 4) for both short- and long-term tests (Figures 9 and 10). The average matched the setpoint values in all tests. The maximum deviation from the average was 0.1 hPa; hence we can specify that the chamber control system is able to assure pressure stability of ±0.1 hPa from the setpoint value.

**Table 1.** Stability of barometric pressure (hPa) during the stages of steady setpoint (stages 2 and 4). The table shows descriptive statistics of the pressure setpoint for both short- and long-term tests.

| Test | Setpoint (hPa) | Min (hPa) | Average (hPa) | Max (hPa) | Standard Deviation (hPa) |
|---|---|---|---|---|---|
| Test 1—stage 2 | 945 | 944.97 | 945.00 | 945.07 | 0.024 |
| Test 1—stage 4 | 955 | 954.91 | 955.00 | 955.07 | 0.034 |
| Test 2—stage 2 | 935 | 934.90 | 934.99 | 935.08 | 0.036 |
| Test 2—stage 4 | 925 | 924.90 | 925.00 | 925.12 | 0.051 |
| Test 3—stage 2 | 945 | 944.98 | 945.00 | 945.04 | 0.024 |
| Test 3—stage 4 | 955 | 954.97 | 955.00 | 955.08 | 0.027 |
| Test 4—stage 2 | 935 | 934.97 | 935.00 | 935.03 | 0.022 |
| Test 4—stage 4 | 925 | 924.96 | 925.00 | 925.02 | 0.026 |

Goyette et al. [24] designed a barometric chamber in which the maximum difference between the instantaneous setpoint pressure and the recorded pressure was 1.1 hPa. According to the authors, their system was able to adequately simulate the range of pressure to which insects are normally subjected in their natural environment. Their control system was equipped with ordinary solenoid valves and an on/off algorithm. They also defined a dead band of ±1.0 hPa in the control algorithm to improve the stability of their system. Although not reported by [24], the conventional solenoid valves used as their actuators to control the pressure should present a short lifespan due to the high number of operations required in entomology experiments, which are usually quite long. The proportional solenoid valves employed in the current design allow the use of better control algorithms and the achievement of higher performance and should also last longer in applications such as the one addressed here. The chamber described here presents a significant contribution in terms of performance since its stability and accuracy were superior to that described by [24]. The main reasons for the better performance of the current chamber are the proportional solenoid valves (which are better than ordinary solenoid valves) and the PI-controller (which is better than an on/off controller). Although the PI-controller was tuned empirically through several attempts, a proper performance was achieved for the entomology applications.

The literature also reports smaller barometric chambers in which pressure was manually controlled by injecting volumes of air and checking a barometric pressure gauge [10,17,29]. Although such designs are technically feasible, they require the user to continuously check the pressure and, despite this, poor performance (i.e., accuracy, stability) is expected to occur while controlling the pressure. In addition, reproducing real pressure fluctuations would hardly be achieved by manual control, since the changes in barometric pressure are small and occur very slowly in nature.

*3.4. Air Tightness*

Figure 11 presents the results of the air tightness tests and indicates that the chamber has leakages of air. In these air tightness tests, the pump was turned on to pressurize/depressurize the chamber and then turned off about 100 s from the start of the test. After turning off the pump, the barometric pressure returned to its initial values after about 100 and 200 s in the positive and negative pressure tests, respectively. Short- and long-term tests demonstrated that the control system is able to reach and maintain the

setpoint pressure although air leakage exists. In addition, perhaps a small level of leakage might be beneficial to enable air renewal within the chamber during the experiments. The system could be improved by installing sensors to monitor gas concentrations (e.g., $O_2$, $CO_2$) inside the chamber.

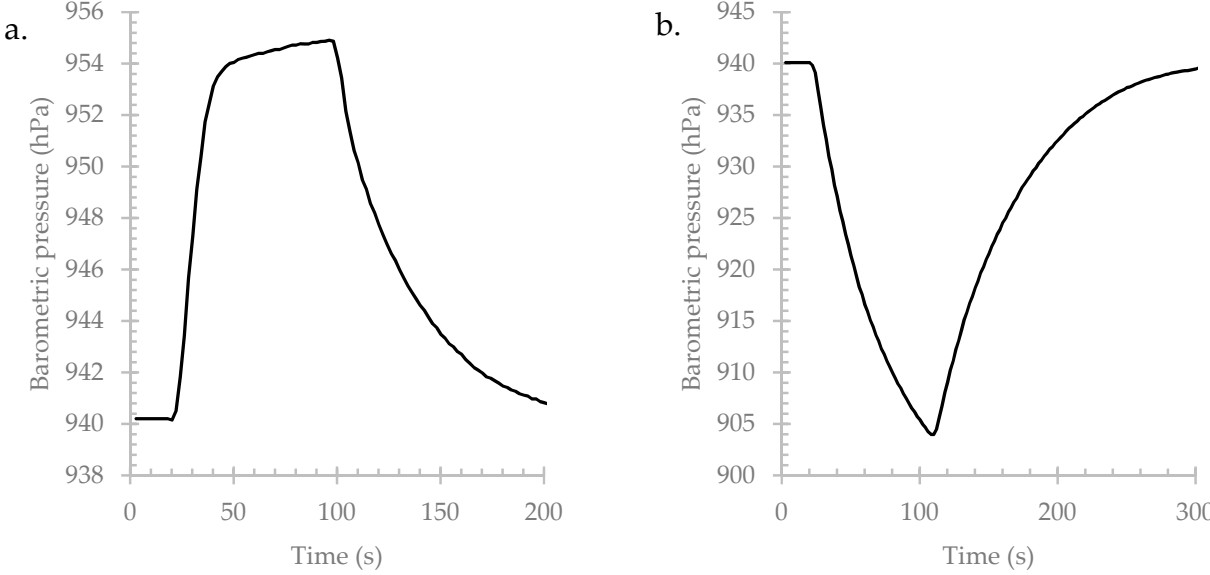

**Figure 11.** Results of air tightness test when operating the chamber with positive (**a**) and negative barometric pressure (hPa) (**b**) to show possible leakages of air in the chamber.

### *3.5. Entomology Experiments: Feeding Activity of Insects*

The number of stylet sheaths is a parameter used for inference of food activity and host preference for hemipterans [28,30]. Figure 12a shows stylet sheath damage to bean pods by *E. heros*, where each violet dot indicates a stylet sheath. The mean values of the number of stylet sheath in *E. heros* for high, stable and low pressure conditions were $4.20 \pm 0.51$, $3.44 \pm 0.32$ and $2.75 \pm 0.28$, respectively. Adult feeding of the *E. heros*, quantified by the number of stylet sheaths, was statistically higher under high pressure conditions compared to the low pressure (degrees of freedom = 2, likelihood ratio = 6.25, *p*-value = 0.0439). Environmental factors act directly on insect development and mortality. Changes in the activities pattern after changes in barometric pressure can be adopted by insects as a survival strategy [9,16]. The increase in stylet sheath under conditions of high barometric pressure suggests an increase in the movement and in the search activity of *E. heros* through a feeding site. In general, high pressure conditions are associated with stable and dry weather, also favoring the movement along the pod. Similarly, for the psyllid, *Diaphorina citri* Kuwayama (Hemiptera: Liviidae), low pressure condition resulted in lower movement of the adults in the citrus plants [10].

Figure 12b shows leaf consumption by *D. speciosa*. The leaf consumption ($cm^2$) per adult for high, stable, and low pressure conditions were $0.586 \pm 0.107$, $0.364 \pm 0.08$, and $0.430 \pm 0.08$, respectively. Statistically, the leaf consumption by *D. speciosa* did not differ between the barometric pressure conditions (degrees of freedom = 2, likelihood ratio = 2.54, *p*-value = 0.2812). In [12], it was reported that barometric pressure did not influence feeding behavior of *Conotrachelus nenuphar* (Herbst) (Coleoptera: Curculionidae) either. Feeding behavior is an essential activity for insects and changes in this activity may compromise their reproductive success [31]. The absence of barometric pressure influence on *D. speciosa* feeding activity could be related to particularities of this insect behavior facing adverse weather conditions. Insects are likely to move toward a more favorable microhabitat under adverse weather conditions. This assumption is true, for example, for locusts, which seek a more protected microhabitat when barometric pressure drops [21]. In addition, it is known

that in the hottest periods of the day, *D. speciosa* adults move from the upper to the lower leaves, stems, and shoots without interrupting their feeding [32].

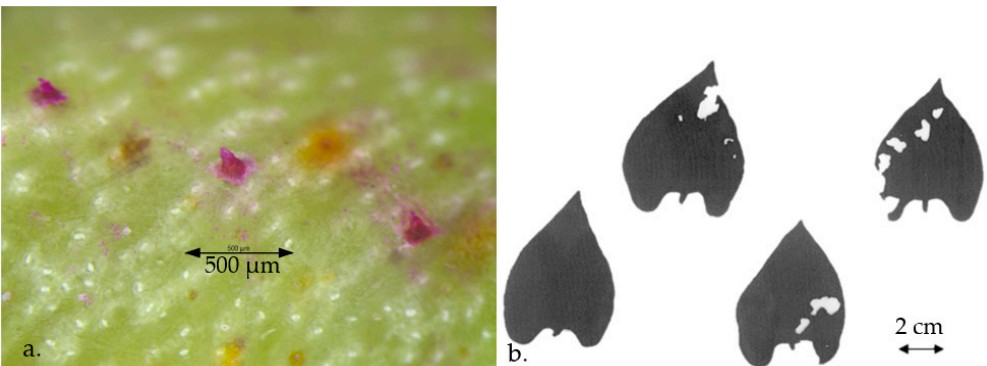

**Figure 12.** Stylet sheath (violet dots) indicates food activity damage to bean pods by *E. heros* (**a**); white holes in the leaves indicate leaf consumption by *D. speciosa* (**b**).

*3.6. Perspectives on Use*

Barometric pressure can affect the response of different living organisms. Control of pressure variation inside a chamber enables the simulation of several pressure conditions that reproduce real conditions. Most studies correlate behavioral patterns with natural barometric pressure variation, while studies of laboratory simulation with pressure control are scarce [33]. Using a barometric pressure chamber in those correlational studies would bring more reliable information about the barometric pressure effects since in natural conditions other abiotic factors such as temperature, humidity, wind, and photoperiod might also influence the individual response. A barometric pressure chamber might be used in future studies focusing on the association of arthropod behavior with the response of plants to simulated adverse weather, which has not yet been elucidated, and/or on arthropod–plant interaction.

Only a few studies have manipulated and controlled barometric pressure using a barometric chamber. In [9], a pressure chamber designed by [24] for insect behavioral experiments was used. Studies [16,23] conducted behavioral experiments inside a controlled pressure chamber at the Advanced Facility for Avian Research (AFAR) at the University of Western Ontario, London, Canada. Study [23] reports that AFAR uses a hypobaric chamber which cannot increase barometric pressure above ambient values. Research work in [10,17] used custom-made manual pressure chambers. As presented here, the design of an automated controlled barometric pressure chamber allows the maintaining, increasing, and decreasing of the barometric pressure, whatever the ambient pressure and without depressurization. The proposed design of the barometric chamber might extend this line of research since no similar equipment can be found in research facilities working with entomology.

**4. Conclusions**

An automated barometric pressure chamber for entomology experiments was designed and evaluated.

Weather conditions in São Paulo state, Brazil, indicated that barometric pressure fluctuations of ±15 hPa from the local barometric pressure are expected to occur. Based on that information, the pressure control system was designed to allow the barometric pressure to be changed by ±15 hPa with respect to the local barometric pressure.

The chamber was designed for entomology experiments consisting of four stages, but other strategies or methodologies could be implemented in the algorithm. In addition, since variations in barometric pressure occur slowly in nature, the control system allows the user to set the rise/fall time (ramp), corresponding to the duration in which a change of pressure will be conditioned. The control system can reproduce slow and rapid barometric pressure

variations due to ordinary or extreme weather conditions. Changes in barometric pressure during ramp stages complied with a linear function, but other mathematical models could be easily implemented. The system can maintain, increase, and decrease the barometric pressure without depressurization.

Short- and long-term evaluations demonstrated that the control system can adjust the target values of barometric pressure and maintain it with a stability of ±0.1 hPa.

The barometric pressure chamber allows phenomena to be filmed while changing the barometric pressure inside it and may be useful to entomology researchers evaluating the influence of abiotic factors on arthropod behaviors and arthropod–plant interactions.

From the entomology experiments carried out for demonstration purposes, we observed that adult feeding of the *E. heros*, quantified by number stylet sheaths, was statistically higher under high-pressure conditions compared to the low pressure. However, the leaf consumption by *D. speciosa* did not differ between the evaluated pressure conditions.

Here we showed that the automated barometric chamber was useful in describing the effect of atmospheric pressure on the feeding activity of phytophagous insects, such as *E. heros* and *D. speciosa*. When combined with other information on the bioecology of these insects, the automated barometric chamber may help to understand their way of life in the natural environment and, in the case of insect pests, assist in their management.

**Author Contributions:** Conceptualization, C.M.C. and J.M.S.B.; methodology, C.M.C., A.P.C. and E.A.d.S.; software, E.A.d.S. and A.P.C.; validation, C.M.C., A.P.C. and E.A.d.S.; formal analysis, C.M.C. and A.P.C.; investigation, C.M.C. and A.P.C.; resources, C.M.C. and A.P.C.; data curation, A.P.C.; writing—original draft preparation, C.M.C. and A.P.C.; writing—review and editing, C.M.C., A.P.C., E.A.d.S. and J.M.S.B.; visualization, C.M.C. and A.P.C.; supervision, A.P.C. and J.M.S.B.; project administration, J.M.S.B.; funding acquisition, J.M.S.B. All authors have read and agreed to the published version of the manuscript.

**Funding:** This research was funded by the São Paulo Research Foundation (FAPESP, grant #2014/140623). This study was supported by INCT—Semiochemicals in Agriculture (FAPESP and CNPq, grants #2014/50871-0 and #465511/2014-7, respectively).

**Institutional Review Board Statement:** Not applicable.

**Informed Consent Statement:** Not applicable.

**Data Availability Statement:** Not applicable.

**Conflicts of Interest:** The authors declare no conflict of interest.

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
