# Peer review of "Automated Barometric Chamber for Entomology Experiments: Arthropods’ Behavior and Insect-Plant Interactions"

_applsci, doi:10.3390/app12146971_

Round 1
Reviewer 1 Report
The manuscript “Automated barometric chamber for entomology experiments: arthropods’ behavior and insect–plant interactions” by Costa et al., is an interesting study for applied ecologists and the Section Ecology Science and Engineering of the Journal is a suitable venue for the publication.
On the list of contributors we do not find the participation of engineering staff involved in the project of designing the automated barometric chamber. May I presume this was designed by entomologist and agriculture staff only?
I’m concerned with the fact that no control experiments were done, and their results contrasted, using “classical” methodological procedures and/or natural environmental under which the barometric pressure were measured.
It is not very clear the practical interest of the studies taking into account the variation of atmospheric pressure in the feeding activity of the phytophagous insects. Are the small alterations in the feeding rate significant for agriculture productivity?
Author Response
The authors appreciate the suggestions presented by the reviewers. All questions and suggestions were addressed.
Reviewer: The manuscript “Automated barometric chamber for entomology experiments: arthropods’ behavior and insect–plant interactions” by Costa et al., is an interesting study for applied ecologists and the Section Ecology Science and Engineering of the Journal is a suitable venue for the publication.
On the list of contributors, we do not find the participation of engineering staff involved in the project of designing the automated barometric chamber. May I presume this was designed by entomologist and agriculture staff only?
Authors: The automated barometric chamber was not designed by entomologist staff. The hardware and software were designed by Antonio P. Camargo and Eric A. Silva, which have expertise on engineering, electronics, and instrumentation. Both authors are not part of the Entomology Department.
Reviewer: I’m concerned with the fact that no control experiments were done, and their results contrasted, using “classical” methodological procedures and/or natural environment under which the barometric pressure was measured.
Authors: In the condition named as “1: stable (S)”, insects were kept at 950 hPa during the whole experiment. This condition served as the control in the experiments. Please see the details presented in section 2.6. The comparison of results obtained in the automated barometric chamber with results obtained in the natural environment was not part of the purposes of this research.
Reviewer: It is not very clear the practical interest of the studies taking into account the variation of atmospheric pressure in the feeding activity of the phytophagous insects. Are the small alterations in the feeding rate significant for agriculture productivity?
Authors: The automated barometric chamber can be used for a wide range of ecological activities involving arthropod behavior and insect-plant interactions in the laboratory. In this manuscript, in addition to describing this equipment in detail, we used the effect of atmospheric pressure on the feeding activity of phytophagous insects to exemplify its use. Other possibilities, for example, have already been demonstrated in the sexual behavior of insects (please see ref. 16, Pellegrino et al. 2013) and the foraging activity of leafcutter ants (please see ref. 25, Sujimoto et al. 2020) with the same barometric camera.
Reviewer 2 Report
This is an excellent study. The degree to which insect behavior may be influenced by barometric pressure fluctuations is intriguing and little is known, most probably because of the difficulty of having laboratory facilities that can produce such fluctuations under controlled conditions. In this paper such a device is described, and also used for making two experiments on how/if feeding behavior is influenced by fluctuations in barometric pressure. Interestingly feeding behavior was influenced in one of the insects studied, the hemipteran Euschistus heros, but not in the other, the coleopteran Diabrotica speciosa.
The manuscript is interesting and exceedingly well written, and I have only one, minor, suggestion for revision, namely to provide the taxonomic affiliation of the two insects when they are mentioned for the first time, on line 99 in the final sentence of the Introduction - in the present form this information is given much later, in the Materials and Methods section on lines 280 - 281.
Author Response
The authors appreciate the suggestions presented by the reviewers. All questions and suggestions were addressed.
Reviewer: This is an excellent study. The degree to which insect behavior may be influenced by barometric pressure fluctuations is intriguing and little is known, most probably because of the difficulty of having laboratory facilities that can produce such fluctuations under controlled conditions. In this paper such a device is described, and also used for making two experiments on how/if feeding behavior is influenced by fluctuations in barometric pressure. Interestingly feeding behavior was influenced in one of the insects studied, the hemipteran Euschistus heros, but not in the other, the coleopteran Diabrotica speciosa.
The manuscript is interesting and exceedingly well written, and I have only one, minor, suggestion for revision, namely to provide the taxonomic affiliation of the two insects when they are mentioned for the first time, on line 99 in the final sentence of the Introduction - in the present form this information is given much later, in the Materials and Methods section on lines 280 - 281.
Authors: Agreed. The taxonomic affiliation was included.
Thank you very much.
Reviewer 3 Report
This device is useful and well-designed. My experience with maintaining a Drosophila lab had me asking right away: where is the oxygen and carbon dioxide monitoring? I think that should be built in by default.
The remote control and precise pressure maintenance will be extremely valuable to research. I think the figures look appropriate but most need a sentence to explain the significance of what we are looking at (in the image description, separate from the general text).
Two thoughts: have you tried with extreme conditions to test your equipment (e.g. very high or very low pressure testing) and can you build in something region-specific for detecting average local barometric pressure?
Line items:
- Paragraph starting on line 46 could really use a figure for visualizing global barometric and weather patterns
- Sentence beginning on line 62 needs a citation as it makes an important claim
- Figure 1 - label the images a and b and the table as c
- Figure 2 - add a sentence explaining briefly what the significance of the graph is, what conclusions it makes
- Figure 3 - I think adding a visual of the flies inside the device will help
- Section 2.3 also needs a visual for the air flow, in my opinion
- Figure 4 - add a sentence explaining briefly what the significance of the image is, what conclusions it makes
- Figure 5 - add a sentence explaining briefly what the significance of the image is, what conclusions it makes
- Figure 8 - add a sentence explaining briefly what the significance of the graphs are, what conclusions, also maybe collapse them into one graph
-Figure 9 - add a sentence explaining briefly what the significance of the image is, what conclusions it makes
- The paragraph starting at line 400 starts with a {24} and you should name the group (formatting opinion; Goyette et al (24) designed...)
- Figure 10 - add a sentence explaining briefly what the significance of the graphs means
- Table 1 also needs a brief explanation of what we are looking at
- Figure 11 - add a sentence explaining briefly what the significance of the graphs are
- Figure 12 - label (a) and (b) on the image and add a brief explanation
- The final part at line 529 needs a wrap-up sentence for style - conclude what the two experiments did and how that ties up the story
Author Response
The authors appreciate the suggestions presented by the reviewers. All questions and suggestions were addressed.
Reviewer: This device is useful and well-designed. My experience with maintaining a Drosophila lab had me asking right away: where is the oxygen and carbon dioxide monitoring? I think that should be built in by default.
Authors: When the automated barometric chamber was built the entomology team was aware of the importance of monitoring O2 and CO2 inside the chamber, but these sensors were not installed at that period because of budget constraints. Installing both sensors to monitor the concentration of O2 and CO2 in the chamber is planned for further investigations using the barometric chamber.
Reviewer: The remote control and precise pressure maintenance will be extremely valuable to research. I think the figures look appropriate but most need a sentence to explain the significance of what we are looking at (in the image description, separate from the general text).
Authors: Additional description was added in figures captions.
Reviewer: Two thoughts: have you tried with extreme conditions to test your equipment (e.g. very high or very low pressure testing) and can you build in something region-specific for detecting average local barometric pressure?
Authors:
- As shown in Figure 8, the equipment was evaluated within limits of local atmospheric pressure +19.63 hPa and -47.15 hPa. Both positive and negative pressures are sufficient for the proposed entomology experiments, which require pressure variations of ±15 hPa. The observed differences in pressure capacity while pressurizing and depressurizing the chamber are related to the operational characteristics of the air pump. Higher values of pressure variation could be obtained by selecting an air pump of higher flow capacity.
- Measuring local barometric pressure could be achieved using a data acquisition system and a good-quality barometric pressure transducer. In fact, sometimes such transducer is part of automatic weather stations.
Reviewer: Line items:
- Paragraph starting on line 46 could really use a figure for visualizing global barometric and weather patterns
Authors: Barometric pressure varies from place to place according to altitude, weather conditions and even weather season. Sorry but presenting the information you suggested, at global scale, is not feasible in this manuscript.
Reviewer: - Sentence beginning on line 62 needs a citation as it makes an important claim.
Authors: References 9 to 22 are all related to that sentence.
Reviewer:- Figure 1 - label the images a and b and the table as c
Authors: Agreed. Figure caption was updated.
Reviewer:- Figure 2 - add a sentence explaining briefly what the significance of the graph is, what conclusions it makes
Authors: Agreed. Figure caption was updated. Figures 1 and 2 indicate the range of values and natural fluctuations of barometric pressure in places of São Paulo state. Both figures support the following requirement stated while developing the barometric chamber: “Based on the historical dataset analyzed, we determined that the barometric chamber must allow the barometric pressure to be changed by up to ±15 hPa with respect to the current barometric pressure. In addition, since variations in barometric pressure occur very slowly in nature, the control system must make it possible to set the rise/fall time corresponding to the duration in which a change of pressure will be conditioned.”
Reviewer: - Figure 3 - I think adding a visual of the flies inside the device will help
Authors: Figure 3 aims to indicate the barometric chamber and its main components. Insects within the chamber are shown in Figure 6.
Reviewer: - Section 2.3 also needs a visual for the air flow, in my opinion
Authors: Sorry but displaying air flow streamlines might require special equipment which are not part of the purposes of this manuscript. Pressure inside the chamber is not influenced by the air flow streamlines. Only the pressure/vacuum pump characteristics is important.
Reviewer: - Figure 4 - add a sentence explaining briefly what the significance of the image is, what conclusions it makes
Authors: Agreed. Figure caption was updated. Figure 4 is a diagram that represents the components of the electrical circuit. All the contents written in section 2.4 are summarized in Figure 4.
Reviewer:- Figure 5 - add a sentence explaining briefly what the significance of the image is, what conclusions it makes
Authors: Agreed. Figure caption was updated. While Figure 4 is a diagram of the electric circuit, Figure 5 is a diagram of the software architecture. All contents written in the section 2.5 are summarized in Figure 5. Figure 4 is a diagram of hardware and Figure 5 is a diagram of software.
Reviewer: - Figure 8 - add a sentence explaining briefly what the significance of the graphs are, what conclusions, also maybe collapse them into one graph
Authors: Agreed. Figure caption was updated. Figure 8 presents the maximum variation in barometric pressure while pressurizing and depressurizing the chamber. During 20 minutes of testing for the positive pressure, the barometric pressure increased by 19.63 hPa, starting from an initial value of 940.41 hPa. For the initial pressure of 940.36 hPa, barometric pressure decreased by 47.15 hPa while evaluating the negative pressure. Both positive and negative pressures are sufficient for the proposed entomology experiments, which require pressure variations of ±15 hPa.
Merging the graphs will difficult visualization of the range of evaluated values of barometric pressure.
Reviewer: -Figure 9 - add a sentence explaining briefly what the significance of the image is, what conclusions it makes
Authors: Agreed. Figure caption was updated. Figure 9 shows the results of short-term tests run to demonstrate that the chamber works as expected. Test 1 (Figure 9a and b) consisted of the following stages: (1) stage 1 –the barometric pressure was gradually increased to 945 hPa in 10 min; (2) stage 2 – a pressure of 945 hPa was maintained for 10 min; (3) stage 3 – the barometric pressure was gradually increased to 955 hPa in 60 min; (4) stage 4 – a pressure of 955 hPa was maintained for 10 min. Test 2 (Figure 9c and d) consisted of the following stages: (1) stage 1 – the barometric pressure was gradually decreased to 935 hPa in 10 min; (2) stage 2 – a pressure of 935 hPa was maintained for 10 min; (3) stage 3 – the barometric pressure was gradually decreased to 925 hPa in 60 min; (4) stage 4 – a pressure of 925 hPa was maintained for 10 min.
The results of barometric pressure obtained in the short-term and long-term experiments (Figures 9 and 10) demonstrate that the control system operated properly, complying with the setpoint values and with the specifications for rise/fall time. The results from test stages 1 and 3 (i.e., ramps) indicated linearity while varying the barometric pressure. The linear function in the ramp stages was chosen by the entomology team as a good way to simulate real fluctuations, but other mathematical models could be implemented to change pressure in these stages.
Reviewer: - The paragraph starting at line 400 starts with a {24} and you should name the group (formatting opinion; Goyette et al (24) designed...)
Authors: Agreed.
Reviewer: - Figure 10 - add a sentence explaining briefly what the significance of the graphs means
Authors: Agreed. Figure caption was updated. Figure 10 shows the results of long-term tests run to demonstrate that the chamber works as expected. Test 3 (Figure 10a and b) consisted of the following stages: (1) stage 1 – the barometric pressure was gradually increased to 945 hPa in 60 min; (2) stage 2 – a pressure of 945 hPa was maintained for 20 min; (3) stage 3 – the barometric pressure was gradually increased to 955 hPa in 180 min; (4) stage 4 – a pressure of 955 hPa was maintained for 20 min. Test 4 (Figure 10c and d) consisted of the following stages: (1) stage 1 – the barometric pressure was gradually decreased to 935 hPa in 60 min; (2) stage 2 – a pressure of 935 hPa was maintained for 20 min; (3) stage 3 – the barometric pressure was gradually decreased to 925 hPa in 180 min; (4) stage 4 – a pressure of 925 hPa was maintained for 20 min.
The results of barometric pressure obtained in the short-term and long-term experiments (Figures 9 and 10) demonstrate that the control system operated properly, complying with the setpoint values and with the specifications for rise/fall time. The results from test stages 1 and 3 (i.e., ramps) indicated linearity while varying the barometric pressure. The linear function in the ramp stages was chosen by the entomology team as a good way to simulate real fluctuations, but other mathematical models could be implemented to change pressure in these stages.
Reviewer:- Table 1 also needs a brief explanation of what we are looking at
Authors: Agreed. Table caption was updated. Table 1 shows descriptive statistics of the stages of the steady setpoint (i.e., stages 2 and 4) for both short- and long-term tests (Figures 9 and 10). The average matched the setpoint values in all tests. The maximum deviation from the average was 0.1 hPa; hence we can specify that the chamber control system is able to assure pressure stability of ±0.1 hPa from the setpoint value. It indicates quality of the control system.
Reviewer:- Figure 11 - add a sentence explaining briefly what the significance of the graphs are
Authors: Agreed. Figure caption was updated. Figure 11 presents the results of the air tightness tests and indicates that the chamber has leakages of air. In these air tightness tests, the pump was turned on to pressurize/depressurize the chamber and then turned off about 100 s from the start of the test. After turning off the pump, the barometric pressure returned to its initial values after about 100 and 200 s in the tests of positive and negative pressure, respectively. Short- and long-term tests demonstrated that the control system is able to reach and maintain the setpoint pressure although air leakage exists. In addition, perhaps a small level of leakage might be beneficial to enable air renewal within the chamber during the experiments.
Reviewer:- Figure 12 - label (a) and (b) on the image and add a brief explanation
Authors: Agreed. Figure caption was updated. Labels (a) and (b) were added to the figure. Figure 12a shows stylet sheath damages to bean pods by E. heros, where each violet dot indicates a stylet sheath. Figure 12b shows leaf consumption by D. speciosa.
Reviewer:- The final part at line 529 needs a wrap-up sentence for style - conclude what the two experiments did and how that ties up the story
Authors: We appreciate the suggestion, and a wrap-up has been included in the place mentioned.
Round 2
Reviewer 1 Report
No additional coments.